# Antioxidant Compounds of Potato Breeding Genotypes and Commercial Cultivars with Yellow, Light Yellow, and White Flesh in Iran

**DOI:** 10.3390/plants12081707

**Published:** 2023-04-20

**Authors:** Somayeh Bahadori, Mousa Torabi Giglou, Behrooz Esmaielpour, Bahram Dehdar, Asghar Estaji, Christophe Hano, Gholamreza Gohari, Marzia Vergine, Federico Vita

**Affiliations:** 1Department of Horticulture, Faculty of Agriculture, University of Mohaghegh Ardabili, Ardabil 56199-11367, Iran; 2Ardebil Agriculture and Natural Resources Center, Agricultural Research, Education and Extension Organization (AREEO), Ardabil 56951-57451, Iran; 3Laboratoire de Biologie des Ligneux et des Grandes Cultures (LBLGC), INRAE USC1328, Université d’Orléans, 28000 Chartres, France; 4Department of Horticulture, Faculty of Agriculture, University of Maragheh, Maragheh 55181-83111, Iran; 5Department of Biological and Environmental Sciences and Technologies, University of Salento, 73100 Lecce, Italy; 6Department of Biosciences, Biotechnology and Environment, University of Bari Aldo Moro, 70121 Bari, Italy

**Keywords:** antioxidant, ascorbic acid, carotenoid, FRAP, soluble proteins

## Abstract

Potatoes are a staple food with high antioxidant properties that can positively affect population health. The beneficial effects of potatoes have been attributed to tuber quality. However, the tuber quality related researches at genetic levels are very few. Sexual hybridization is a powerful strategy for producing new and valuable genotypes with high quality. In this study, 42 breeding potato genotypes in Iran were selected based on appearance characteristics such as shape, size, color, eyes of tubers, and tuber yield and marketability. The tubers were evaluated for their nutritional value and properties, viz. phenolic content, flavonoids, carotenoids, vitamins, sugars, proteins, and antioxidant activity. Potato tubers with white flesh and colored skin had significantly higher levels of ascorbic acid and total sugar. The result showed that higher phenolic, flavonoid, carotenoid, protein concentration, and antioxidant activity were noted in yellow-fleshed. Burren (yellow-fleshed) tubers had more antioxidant capacity in comparison to genotypes and cultivars, which did not differ significantly with genotypes 58, 68, 67 (light yellow), 26, 22, and 12 (white). The highest correlation coefficients in antioxidant compounds were related to total phenol content and FRAP, suggesting that phenolics might be crucial predictors of antioxidant activities. The concentration of antioxidant compounds in the breeding genotypes was higher than in some commercial cultivars, and higher antioxidant compounds content and activity were detected in yellow-fleshed cultivars. Based on current results, understanding the relationship between antioxidant compounds and the antioxidant activity of potatoes could be very helpful in potato breeding projects.

## 1. Introduction

Potato (*Solanum tuberosum* L.) is of the most common nutrient sources after iconic crops such as rice, wheat and maize [1], constituting the significant rate of the food and nutritional demand of human beings [2]. Concerning available phytochemicals, the relevant crop is a source rich in vitamins A, B, and C, carbohydrates, patatin storage proteins, fiber, antioxidant compounds (for example, caffeic acid), calcium, potassium, phosphorus, and iron [3,4,5,6]. However, the contents of those phytochemicals exhibit critical variations according to the potato cultivars [7]. Of the potato plants, potato tubers are the sources of health-promoting compounds [8]. Antioxidant activities of potatoes have been assayed in very many studies [7,9,10,11,12]. In potato tubers, polyphenols [7,13], ascorbic acid, carotenoids [14], and protein [8] are significant antioxidants. However, of those antioxidants, phenolic acids are the major antioxidant compounds, as common in other crops, as well [15,16]. Concerning phenolics available, chlorogenic acid, gallic acid, protocatechuic acid, and caffeic acid are phenolics identified in potato skins [10]. On the other hand, rutin, quercetin, myricetin, kaempferol, and naringenin are important flavonoids in potatoes [17]. Flavonoids in the flesh and skin cause pigmentation of cultivated potato varieties [16]. Subsequently, colored potato genotypes have significant antioxidant activity [18].

The flesh color of potato tubers is related to the accumulation of two different classes of pigments. For instance, the accumulation of anthocyanins results in red, blue, or purple colors of the flesh and white, yellow, and orange flesh due to carotenoid levels [19]. Carotenoids are tetraterpens, and their isoprenoid structure consists of 40 carbons; they have a long chain of double bonds [12] and exhibit similar functions to vitamin A activity [20,21]. Carotenoids are powerful antioxidants to scavenge free radicals due to their double bonds [15]. Due to the importance of carotenoids for health [22], considerable attention has been paid to the selection and development of food products by increasing the concentration of total and individual carotenoids [6]. Concerning carotenoid compounds, violaxanthin, antheraxanthin, lutein, and zeaxanthin are well-known in potatoes [23,24], but their concentrations critically vary according to potato varieties [19].

Of the vitamins available, Vitamin C is the most abundant in potato tuber, with a range of amounts from 10 to 40 mg per 100 g of fresh weight [18]. As a powerful antioxidant, Vitamin C exhibits significant functions such as protective roles against oxidative stress and involvement in a plethora of cellular activities (i.e., cell division and growth, organogenesis, and collagen synthesis) [25]. In addition, protein constitutes one-third of the dry weight of the potato tuber and is composed of three major groups: patatin, protease inhibitors, and other proteins [26]. Among those major groups, patatin, as a unique potato tuber protein, suppresses/inhibits free radicals [6,27]. The potato proteins have been classified as “excellent” (90 out of 100) for their biological values [28]. In addition, patatin proteins have desirable functional properties, such as foam formation and stabilization, fat emulsion, or gelling [29].

Carbohydrates, especially starch, constitute a large portion of potato tubers [29]. After processing the potato tubers at high temperatures, asparagine (amino acids), glucose, or fructose (reducing sugars) can lead to the formation of acrylamide. Subsequently, acrylamide might cause undesirable browning or bitter taste [30,31]. In this regard, the content of sugar is of the critical parameters to be considered in the selection of new cultivars [29].

The production or introduction of promising potato cultivars with desired phytochemicals is of the major targets of agricultural systems. In this regard, some breeding genotypes and commercial cultivars were screened for their phenolics, flavonoids, ascorbic acid, carotenoids, proteins, and sugars, as well as antioxidant activities (DPPH and FRAP). Furthermore, the antioxidant activities were correlated with the phytochemicals available.

## 2. Results and Discussion

### 2.1. Total Flavonoids and Phenolic Content (TFC, TPC)

TPC of yellow- or light-yellow-flesh potatoes ranged from 48.6 to 96 µg mg^−1^ FW, whereas it ranged between 38 to 92 µg mg^−1^ FW in white-flesh potatoes (Table 1). Of the cultivars, “Burren” had the highest content of phenolics (96 µg mg^−1^ FW). On the other hand, the TFC of the cultivars varied between 1.32 and 4.6 µg of quercetin equivalent per mg FW (Table 1). The findings of the present study are higher than cultured potato tubers’ phenolic acid and flavonoid content [32]. In addition, the colored potato cultivars had twice the content of phenolics in relation to the white skin potato cultivars. As in the case of TPC, the cultivar “Burren” also had the highest TFC (4.6 µg mg^−1^ FW). Regarding their significance levels, the cultivar “Burren” was not statistically separated from cultivars 58, 26 and 68. However, the cultivar “Burren” was statistically different from other cultivars in terms of TFC (*p* ˂ 0.01).

Overall, yellow-flesh potatoes had more TPC and TFC than light yellow- and white-flesh genotypes. For example, the lowest amount of TFC (genotype 27) and TPC (genotype sp) was observed in white-flesh potatoes (Table 1). The flavonoids and phenolic acid content in purple or red flesh cultivars were twice and three to four times higher than in white-fleshed cultivars [8,9,32]. Such a high variation can be explained by the fact that pigmented potatoes have a high level of chlorogenic acid isomers compared to non-pigmented potatoes. Subsequently, high levels of chlorogenic acid and its isomers, as well as caffeic acid in red and purple potatoes, are manifested into high antioxidant activity [14]. Overall, very significant differences were noted between potato genotypes or cultivars with respect to the TPC and TFC (*p* ˂ 0.01). A plethora of reports also uttered that phenolic compounds were very variable according to the genotypes/cultivars [7,9,10,11,12] and the associated flesh color [33,34]. Such differences have been attributed either to phenolic compounds’ synthesis levels or the diversification of the compounds [10]. In addition to the effects of environmental phenomena, the level of antioxidant activity of the crop is affected by genotype, field conditions, soil type, plant growth stage, and storage conditions [18].

### 2.2. Antioxidant Activity (DPPH and FRAP)

With respect to estimating the antioxidant activities of samples, there are many methods applied in potato samples (i.e., DPPH, FRAP, and ABTS) [7,9,11,35]. Each method has a different mechanism to remove, reduce or inhibit free radicals. For that reason, more appropriate results might be achieved with the use of at least two methods [11,15]. For that reason, potato cultivars were assayed for their potential antioxidant activities using two methods, namely DPPH (2,2-diphenyl-1-picrylhydrazyl scavenging activity) and FRAP (Ferric Reducing Antioxidant Power). Accordingly, FRAP values and DPPH radical inhibitory activities ranged from 31.7 to 89.4 µg mg^−1^ FW and from 27.9 to 73.5%, respectively (Table 1). Among the three color groups of potato flesh, white-flesh (genotypes 27, sp, 35, and 39) potatoes had the lowest amount of DPPH radical inhibitory activity, and FRAP means (31.7 µg mg^−1^ FW) in white-flesh potatoes (genotype SP). At the same time, yellow-flesh potatoes had the highest amount of DPPH radical inhibitory activities and FRAP. Even though yellow-flesh potatoes have a higher antioxidant capacity than white flesh, genotype 68 with white flesh has a higher antioxidant capacity (DPPH radical inhibitory activities) than some yellow or light yellow flesh potatoes.

In contrast, most yellow flesh samples had higher activity than white flesh samples. The observed differences in FRAP values were statistically significant, and the highest value was found in the cultivar “Burren” (yellow flesh). However, the high value in “Burren” was not statistically different from other potato cultivars (Table 1).

High antioxidant activity has been reported in purple flesh and skin potatoes, purple flesh potatoes have more DPPH radical scavenging activity than yellow potatoes [7,9,11]. Anthocyanins and phenolic acids are considered the main antioxidant compounds in potatoes [9,36]. Chlorogenic acid, gallic acid, caffeic acid, and catechins are the main contributors to the antioxidant capacity of white or yellow flesh potatoes. In contrast, in potatoes with purple and red flesh, anthocyanins are common agents for antioxidant activity [14].

### 2.3. Total Carotenoids Content (TCC)

Table 1 about total carotenoids showed that the carotenoid content obtained from genotype 6 and other genotypes was significantly different (*p* ˂ 0.01; Table 1). However, the highest amount of TCC was reported in genotype 6 (yellow flesh) Table 1. The tubers with yellow flesh and skin were high in carotenoids, and the results of our experiments are consonant with the findings of other studies [12]. Statistical analysis showed that darker yellow flesh has the highest average carotenoids, which are not significantly different from light yellow flesh. Carotenoid content is significantly lower in potatoes with white flesh. However, the amount of potato carotenoid with white flesh and pink skin (Fontane and genotypes 40 and 11) does not differ from the types of yellow or light yellow flesh (genotypes 26, 42, and 7, cultivars Jelly and Burren).

The color of potato tubers is affected by the concentration and composition of carotenoids. Carotenoid concentration significantly correlates with the yellow color of tuberous flesh [6]. In several studies, tuber carotenoid content and type have been measured and identified in a wide range of potato germplasm. In crosses of *Solanum stenotomum* and *Solanum phureja*, a population of diploid progeny was obtained with white, yellow, and orange flesh colors [19]. Orange flesh contains zeaxanthin; depending on the concentration of zeaxanthin, a dark orange-yellow color appears in the flesh of the tuber [16,37]. Zeaxanthin and lutein are lipophilic compounds made from the precursor of isoprenoid in the plastid, the most abundant carotenoids found in yellow and orange flesh potatoes [6]. The yellow flesh trait results from a dominant allele (*Chy2* allele3) of beta-carotene hydroxylase. The other 10 alleles of *Chy2* do not affect the character of yellow flesh [38].

### 2.4. Total Ascorbic Acid (TAA)

The concentration of TAA varied from 10.2 to 34.3 µg mg^−1^ FW, being significantly different between 53 genotypes and cultivars (Table 1). The content of vitamin C in tubers with colored skin and white flesh was higher than in those with light yellow flesh. The highest levels of TAA were observed in genotype 387 (Table 1). After that, the highest amount of vitamin C was obtained by genotypes 47, 57, 14, 32 (white), 12 (light yellow), 49, 30 (yellow), 54 (white), 7 (yellow), sp (white), 61 (yellow), and 42 (light yellow), while the lowest amount of TAA was recorded in genotype 15 (light yellow). In general, the highest concentration of TAA was found first in tubers with white-fleshed and colored skin, then white-fleshed and yellow-skinned, light yellow-fleshed and colored skin, yellow-fleshed and yellow-skinned, and light yellow-fleshed and yellow-skinned (Table 1).

Potatoes are a valuable source of ascorbic acid, so a 148 g uncooked potato provides 45% of the average adult daily value of ascorbic acid [39]. Fresh potato tubers contain up to 50 mg 100 g^−1^ FW of ascorbic acid, but the amount is reduced to 10 mg 100 g-1 FW during storage time. A cultivar that stores a large amount of ascorbic acid during storage is of value because the tubers are usually stored before consumption [40]. Vitamin C content ranged from 15 and 25 mg 100 g^−1^ FW of potato North American variety and breeding line [13]. Six European cultivars and 27 breeding lines of potato genotypes were grown in each of the 2 years at three different European geographical sites, and the concentration of vitamin C varied between 0.65 to 1.54 mg 100 g^−1^ FW [25]. Significant amounts of ascorbic acid were detected in 12 potato cultivars with yellow-fleshed and red-fleshed, ranging from 88.6 to 282.0 mg 100 g^−1^ FW. A red-fleshed cultivar showed the highest ascorbic acid concentration [41,42]. In addition, high heritability has been reported for vitamin C content [16].

### 2.5. Total Soluble Sugars (TSS)

Table 1 shows that the total sugar contents in potato tubers with light yellow or light yellow and white flesh are between 2.45–36.10 and 2.58–59.73 µg mg^−1^ FW, respectively. The highest amount of total sugar in white flesh (59.73) was observed in the “Fontane” cultivar, and this content was generally higher in white flesh and colored skin than in light yellow and yellow flesh. After Fontane, the highest amount of total sugar was observed in Milva cultivars, genotypes 8, 11, and 47, significantly different from other cultivars and genotypes. In addition, the total sugar concentration in the samples with light yellow flesh (2.45–12.85) was lower than in the potatoes with yellow and white flesh. The lowest sugar content was observed in genotypes 15, 30, and 48, and cultivar Burren.

The total sugar content in potato tubers was reported from 10.30 to 47.10 mg g^−1^ [29]. The results of an experiment showed that the total sugar concentration in potato tubers varies between 7.50 and 74.10 mg g^−1^ DW. The most common type of sugar found in tubers is maltose, which is the reason for the sweet taste of potatoes [43]. Potatoes with high starch and low sugar usually have better cooking and processing quality than potatoes with low starch and high sugar content [44]. The quality of processed potatoes depends on the activity of sucrose synthetase, which also determines yield and starch content in potato tubers [45]. Well-processed cultivars are more capable of removing sucrose during growth completion [30]. The different ability of clones to form reducing sugars was probably due to vacuolar invertase activity, sucrose compartmentalization, or other, by the competing pathways of starch synthesis and degradation [46]. Even if glucose and fructose are produced in moderation, glucose levels are high during growth and storage at 4 °C than potatoes stored at different temperatures [47].

### 2.6. Total Soluble Protein (TSP)

The concentration of total protein was observed in yellow flesh (117–241 µg mg^−1^ FW), light yellow flesh (86–273 µg mg^−1^ FW), and white flesh (64.19–289 µg mg^−1^ FW) tubers, respectively. However, the protein content of potatoes with light yellow flesh (genotypes 26, 22, and 12) was not significantly different from yellow flesh (Burren and Jelly; Table 1).

Soluble proteins make up about 75% of the total protein in potatoes, while approximately 25% of all potato proteins are insoluble proteins that make up the cell wall of potatoes [35]. Previous research assessed the concentration and distribution of TSP for 20 fresh and stored potato tubers. Protein concentrations were higher in the periderm (38 to 73 mg g^−1^ DW) compared to the cortex and pith (30 to 49 mg g^−1^ DW). After 6 months of tubers storage, the protein content of 11 cultivars did not change, but some cultivars showed a decrease and increase in protein content [48].

### 2.7. Correlation

Since antioxidant compounds are the strongest and final indicator to separate in some genotypes and cultivars of potato, several multivariate analyses were performed on the data matrix of genotypes and cultivars to evaluate the antioxidant compounds of genotypes and cultivars of potato and establish the range of responses detectable in this potato collection. Based on cluster analysis, clusters of some genotypes and cultivars of potato could be identified largest distance on the dendrogram (Figure 1). Examination of antioxidant compounds in genotypes and cultivars of potato data obtained from the heatmap showed that the highest TSF was related to genotype 68, which is consistent with the average comparison results (Table 1). Also, the result showed that the highest TPC was related to genotype 58 (Figure 1), which is not consistent with the average comparison results (Table 1).

According to the result of the Pearson correlation coefficient (PCC) in antioxidant compounds of some genotypes and cultivars of Potato (Table 2), the highest Pearson correlation coefficient (PCC) between TPC was related to FRAP. Also, the lowest correlation in antioxidant compounds was associated with TSS and TAA. The result showed that the highest correlation value occurs between DPPH and TFC (0.947).

The analysis of relationships occurring between variables (analyzed compounds) and observations (sample) was deepened, according to Principal component analysis (PCA). By looking at the data reported in Figure 2, we may observe as the first two axes accounted for 92.56% of the total variance of the system (91.79% PC1, 0.86% PC2), thus indicating that samples can be grouped in three different clusters according to their similarities. Genotypes and cultivars, such as 58, 23, 46, and so on, are placed in cluster A due to their similarities (Figure 2).

## 3. Materials and Methods

### 3.1. Plant Material

During 2012, crossbreeds were made between commercial cultivars (Khavaran, Satina, Kayzer, Javid, Agria, Milva, Jelly, Maradona, Impala, Oshina, Burren, Fontane), Stbrkaz and Stbrp clones and clones that are in the cultivar introduction stage (*S. andigena*, *S. stoloniferum,* and *S. tuberosum*). In 2019, 42 advanced clones were obtained from successive selections, and 11 commercial clones were planted in the Ardabil Agricultural Research Center research farm. The image of some of these selected genotypes and commercial cultivars with different skin and flesh colors is shown in Figure 3 and Figure 4.

After harvest, 3 tubers from each clone replicate were selected, cleaned, peeled manually, soaked with liquid nitrogen, and frozen at −80 °C until analyzed.

### 3.2. Measurement of Flavonoids Content

The total flavonoid content was determined by the colorimetric method of aluminum chloride [49,50,51]. In this method, 500µL of standard solution or extract, 2 mL H_2_O, and 150 µL NaNO_2_ 5% are added to a Falcon and mixed for 5 min. Then, 150 µL of 10% AlCl_3_·6H_2_O was added to the mixture solution, and after 5 min, 1000 µL of 1M NaOH was added. After keeping the samples at 24 °C for 15 min, the absorbance of the samples was measured at 415 nm. According to the above-mentioned method, quercetin was used as standard at different concentrations (20–1000 μg mL^−1^). Finally, total flavonoid concentration was calculated as quercetin equivalent on a basis µg mg^−1^ FW.

### 3.3. Determination of Phenolic Concentration

Total phenol content was determined by the Folin–Ciocalteu reagent [7,52,53,54]. Briefly, 0.1 g frozen tuber tissue was homogenized with 1 mL of 80% methanol (*v*/*v*) and then centrifugated at 12,000 rpm for 20 min at 4 °C. Subsequently, 40 μL of the 5-fold diluted extract was added to 1.5 min/mL of the 10-fold diluted Folin-Ciocalteu reagent and mixed thoroughly. The absorbance of the reaction solution was recorded at 725 nm. Gallic acid solutions were prepared in different concentrations (20–1000 μg mL^−1^) and used as a standard curve.

### 3.4. Assay of DPPH Radical Scavenging Activity

The antioxidant capacity of potato flesh extract was measured using the DPPH assay (1,1-DiPhenyl-2-Picryl-Hydrazyl), as described [7,55]. First, DPPH (100 μM) radical solution was prepared in methanol, and 40 μL of the 5-fold diluted supernatant was mixed with 3 mL of DPPH radical solution. After keeping the prepared solution in the dark for half an hour at 24 °C, the reading was performed at 415 nm. DPPH scavenging activity was expressed by inhibiting (percentage) DPPH absorbance.

### 3.5. Assay of Ferric-Reducing Antioxidant Power (FRAP)

Ferric-reducing antioxidant reagent was determined according to the method of [56] by adding 20 mM FeCl_3_·6H_2_O, 10 mM TPTZ (2,4,6-tri (2-pyridyl)-s-triazine) and 0.3 mol/L sodium acetate buffer (pH 3.6) with ratio 1:1:10. The FRAP reagent solution was heated at 37 °C for 10 min. In summary, 60 μL of the extract was pipetted to 3 mL of the FRAP reagent. The samples were then dark incubated for a half-hour, and then the absorbance was recorded at a wavelength of 593 nm. Ferrous sulfate heptahydrate solution was used for the standard curve.

### 3.6. Determination of Carotenoid Content

Carotenoids were performed according to Tang et al. [15] method. Briefly, 5 mL of ethanolic butylated hydroxyl toluene (ethanol/BHT—100:1, *v*/*w*) was added to 0.5 g of the potato flesh. Then, samples were vortexed and heated at 85 °C for 5 min. Subsequently, 500 µL of 80% KOH was added, and samples were vortexed again before placing them in a water bath at 85 °C for 10 min. Then, samples were cooled in an ice-water bath, and 3 mL of cold deionized water was added. After that, 3 mL of N-hexane was added to the mixture and centrifuged (7500 rpm, 5 min). The yellow top phase was transferred to a new falcon tube. Next, N-hexane was added 4 more times every time before sample centrifugation. Samples absorbance was read at 450 and 503 nm using N-hexane as blank.

### 3.7. Determination of Ascorbic Acid (AA)

The ascorbic acid concentration was measured with the di nitro phenyl hydrazine (DNPH) method [57,58]. Briefly, 1 g of fresh flesh potato was properly homogenized with 4 mL of 6% metaphosphoric acid. After centrifugation (15,000 rpm, 15 min, 4 °C), 400 μL of the extract was added to 50 µL of 0.2% 2,6-dichlorophenolindolphenol (DCIP), and the reaction mixtures were kept for 1 h at 24 °C. Then, 1 mL of thiourea 2% (*w*/*v*) and 500 µL of DNPH 2% were poured into the mixture. Subsequently, samples were incubated in a water bath (60 °C, 3 h) before being cooled in an ice bath. Finally, 2.5 mL of 85% H_2_SO_4_ was slowly added to the falcons containing the reaction mixture. The sample absorbance was read at 540 nm against blank.

### 3.8. Determination of Soluble Sugars

Total soluble sugars (TSS) were extracted by adding 1000 µL of 95% ethanol to 100 mg of fresh sample. The supernatant was then transferred into another falcon, and 1 mL of 70% ethanol was added twice to the insoluble fraction. Next, falcons containing 3 mL of supernatant were centrifuged, and 100 μL of extract of supernatant was transferred in a new falcon tube by adding 3 mL of anthrone. Samples were then placed in a water bath at 100 °C for 10 min. The amount of total soluble sugar was determined at 625 nm [59].

### 3.9. Determination of Total Protein Content

Briefly, 0.1 g of the sample was homogenized with 3 mL of buffer phosphate pH 7.8. Then the samples were centrifuged at 4 °C for 20 min at 12,000 rpm, and sample aliquots were read at 595 nm. Bovine albumin serum (BSA) was used as a standard reference. Protein contents were calculated according to Bradford [60].

### 3.10. Statistical Analysis

Data were statistically analyzed according to a 1-way ANOVA test. The significant differences were then evaluated using Duncan’s posthoc test with a significance level of *p* < 0.05. Pearson correlation coefficients were calculated for all the analyzed genotypes. Data from chemical profile and antioxidant capacity were also analyzed using a PCA (principal component analysis), and the results were graphically processed to highlight the contribution of each variable respectively (analyzed compounds) in the samples’ differentiation. All computations were analyzed by SAS 9 software (SAS Institute, Cary, NC, USA).

## 4. Conclusions

In conclusion, this study better explains the relationship between antioxidant activity and the compounds in potatoes with different flesh colors. These data provide opportunities for the whole scientific community to further increase the content of phenolics, flavonoids, and antioxidant capacity by breeding, mainly in white flesh potatoes. However, our data indicated that some potato genotypes with light yellow flesh (genotypes: 26, 22, 12) and white flesh (genotypes: 68, 58, 67) had significant differences in phenolic, flavonoid, carotenoid, protein, and antioxidant activity. The results of the current experiment could be helpful for potato breeders and, eventually, commercial potato producers, giving them new opportunities to promote the production of promising potato cultivars with enhanced levels of high nutrient value. Therefore, potato genotypes rich in phytochemicals represent an important natural source of antioxidants for various applications and products with potentially beneficial effects on human health.

Emerging research perspectives suggest that polyphenolic components are key food constituents involved in preventing chronic diseases. Therefore, the use of phytochemicals may reduce inflammation and the risk of cancer, cardiovascular disease, and diabetes.

Thus, increased knowledge of phenolic components and antioxidant activity of different potato cultivars will lead to greater awareness by the food industry and consumers regarding potatoes as “functional foods” and possibly change food industry practices and consumer habits regarding utilizing specific “high antioxidant” potato cultivars.

## Figures and Tables

**Figure 1 plants-12-01707-f001:**
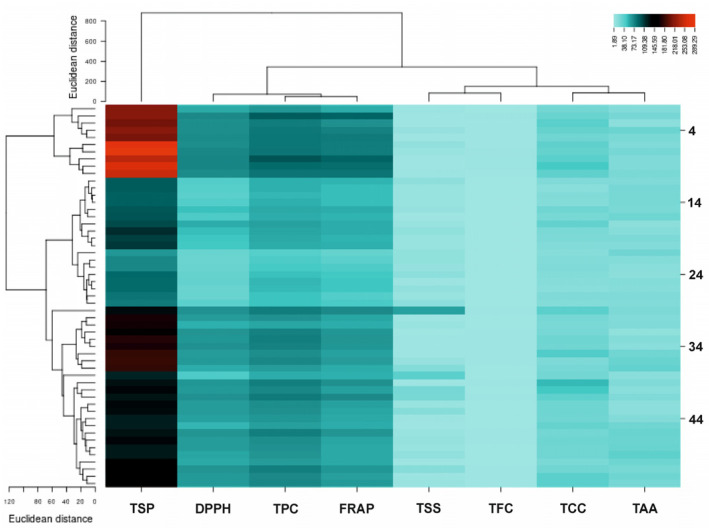
Heatmap of average antioxidant Compounds in Genotypes and Cultivars of Potato. LSD—Least significant difference at *p* < 0.05.

**Figure 2 plants-12-01707-f002:**
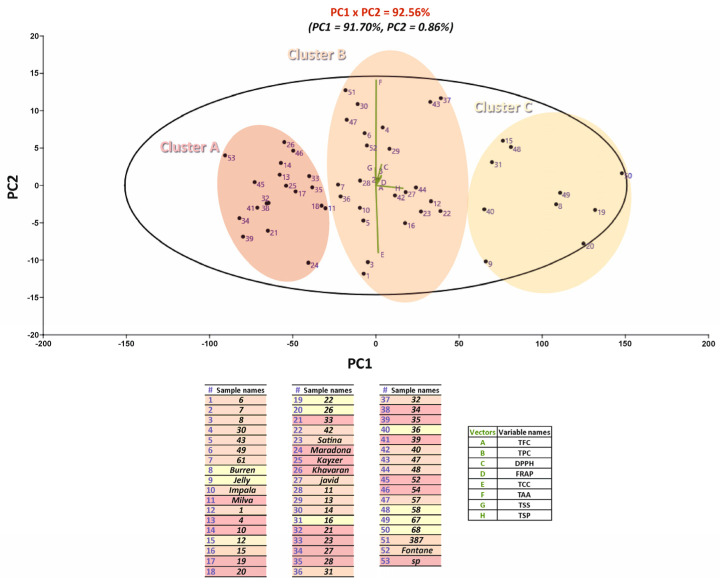
Principal component analysis (PCA) of the chemical profile and antioxidant capacity related to the 53 breeding potato genotypes and cultivars. Variance of component 1 (PC1) = 91.70%, and component 2 (PC2) = 0.86%.

**Figure 3 plants-12-01707-f003:**
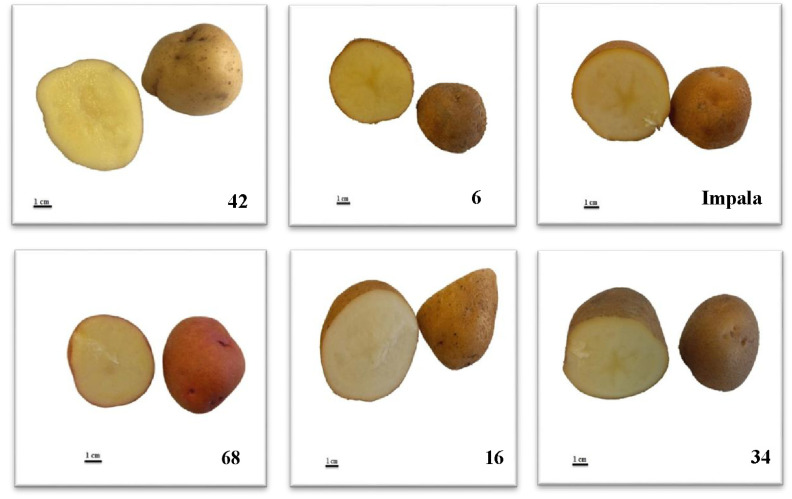
Tuber flesh color of genotypes (42, 6, 68, 16, and 34) and Impala cultivar.

**Figure 4 plants-12-01707-f004:**
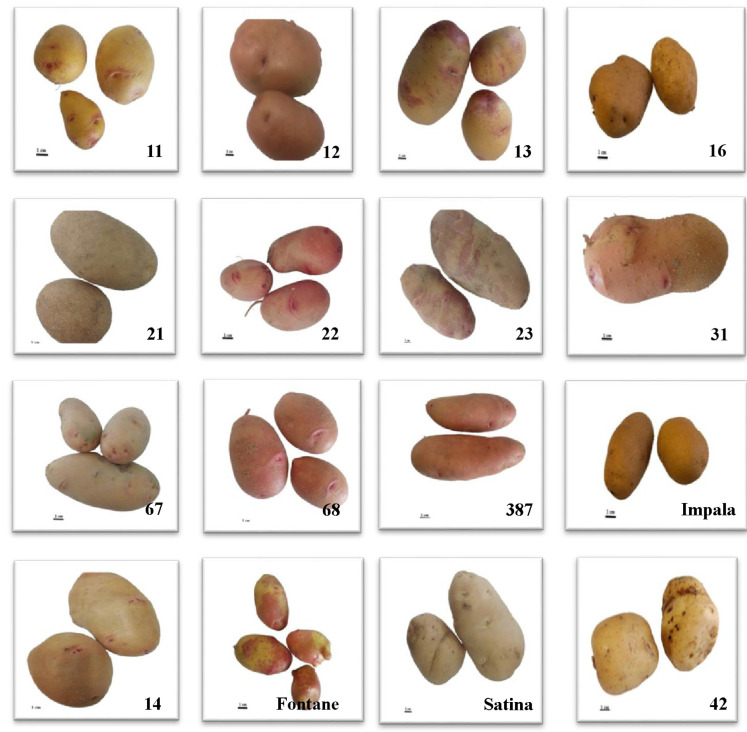
Tuber skin color of genotypes (11, 12, 13, 16, 21, 22, 23, 31, 67, 68, 387, 14, and 42) and Impala, Fontane, and Satina cultivars.

**Table 1 plants-12-01707-t001:** TFC, TPC, DPPH, FRAP, TCC, TAA, TSS and TSP of 53 genotypes and cultivars of potato. Results from Duncan’s posthoc test were reported in the table.

Sample	F/S Colour	TFC (µg mg^−1^ FW)	TPC (µg mg^−1^ FW)	DPPH (%)	FRAP (µg mg^−1^ FW)	TCC (µg mg^−1^ FW)	TAA (µg mg^−1^ FW)	TSS (µg mg^−1^ FW)	TSP (µg mg^−1^ FW)
6	Y/Y	2.80 ^c–i^	75.40 ^a–g^	63.90 ^a–d^	67.20 ^a–i^	50.20 ^a^	18.30 ^d–m^	3.07 ^n–q^	127.70 ^g–n^
7	Y/Y	2.53 ^c–j^	70.70 ^a–i^	62.80 ^a–d^	60.80 ^a–k^	37.43 ^b–e^	26.83 ^a–f^	4.34 ^l–q^	140.30^f–m^
8	Y/Y	2.43 ^d–l^	71.90 ^a–h^	62.90 ^a–d^	60.00 ^a–k^	44.52 ^ab^	14.34 ^i–m^	23.90 ^c^	133.80 ^g–n^
30	Y/Y	2.14 ^e–o^	61.90 ^c–k^	57.80 ^a–g^	57.42 ^c–k^	20.66 ^j–r^	27.16 ^a–e^	2.95 ^n–q^	148.30 ^e–l^
43	Y/Y	2.42 ^d–m^	68.30 ^b–j^	62.00 ^a–e^	59.20 ^a–k^	26.50 ^g–l^	13.30 ^j–m^	5.31 ^j–q^	133.10 ^g–n^
49	Y/Y	2.48 ^d–k^	68.80 ^b–i^	62.10 ^a–e^	58.30 ^b–k^	26.98 ^g–k^	27.51 ^a–e^	7.61 ^h–k^	133.80 ^g–n^
61	Y/Y	2.05 ^f–o^	60.40 ^c–k^	51.14 ^c–k^	55.60 ^c–k^	30.91 ^e–i^	25.21 ^a–h^	3.08 ^m–q^	120.90 ^g–n^
Burren	Y/Y	4.61 ^a^	96.80 ^a^	73.50 ^a^	89.40 ^a^	36.81 ^c–e^	18.60 ^d–m^	2.82 ^o–q^	241.30 ^a–d^
Jelly	Y/Y	3.01 ^c–e^	76.60 ^a–f^	68.62 ^ab^	67.40 ^a–i^	36.91 ^c–e^	12.83 ^k–m^	3.36 ^l–q^	205.80 ^b–f^
Impala	Y/Y	2.40 ^d–m^	67.30 ^b–j^	61.98 ^a–e^	58.90 ^a–k^	26.51 ^g–l^	13.76 ^i–m^	15.53 ^de^	131.40 ^g–n^
Milva	Y/Y	1.91 ^h–o^	55.60 ^d–k^	41.56 ^h–n^	54.33 ^c–k^	25.03 ^h–o^	14.43 ^i–m^	36.1 ^b^	117.60 ^g–n^
1	LY/Y	2.91 ^c–h^	75.80 ^a–g^	64.63 ^a–d^	65.20 ^a–j^	27.07 ^g–k^	16.53 ^g–m^	3.40 ^l–q^	173.00 ^d–h^
4	LY/Y	2.53 ^c–j^	51.30 ^e–k^	36.10 ^k–n^	49.00 ^e–k^	18.26 ^l–r^	21.83 ^c–l^	5.64 ^j–p^	91.10 ^k–n^
10	LY/Y	1.72 ^j–o^	53.70 ^d–k^	38.70 ^i–n^	49.50 ^e–k^	16.25 ^o–r^	21.71 ^c–l^	5.69 ^j–o^	90.90 ^k–n^
15	LY/Y	2.75 ^c–i^	74.38 ^a–h^	65.15 ^a–d^	63.60 ^a–j^	23.00 ^i–q^	10.26 ^m^	2.454 ^q^	157.30 ^e–k^
19	LY/Y	2.00 ^g–o^	57.44 ^d–k^	48.40 ^d–k^	54.50^c–k^	26.19 ^g–m^	20.72 ^c–l^	6.45 ^i–m^	95.70 ^i–n^
20	LY/Y	2.03 ^f–o^	58.58 ^c–k^	49.60 ^c–k^	54.60 ^c–k^	21.38 ^j–r^	14.91 ^i–m^	9.63 ^gh^	112.40 ^g–n^
26	LY/Y	4.20 ^ab^	86.10 ^a–c^	73.17 ^a^	84.20 ^a–c^	44.04 ^a–c^	18.72 ^d–m^	3.4 ^l–q^	261.40 ^ab^
42	LY/Y	2.47 ^d–k^	68.30 ^b–j^	62.08 ^a–e^	59.70 ^a–k^	39.96 ^b–d^	25.14 ^a–h^	2.53 ^pq^	180.90 ^c–g^
Satina	LY/Y	2.80 ^c–i^	74.39 ^a–h^	66.13 ^a–c^	63.70 ^a–j^	26.20 ^g–m^	14.09 ^i–m^	3.36 ^l–q^	166.90 ^e–j^
Maradona	LY/Y	2.07 ^f–o^	59.65 ^c–k^	54.68 ^b–i^	54.80 ^c–k^	32.83 ^d–h^	12.23 ^l–m^	5.09 ^k–q^	101.53 ^e–j^
Kayzer	LY/Y	1.87 ^i–o^	53.93 ^d–k^	38.90 ^i–n^	53.10 ^d–k^	18.90 ^k–r^	18.46 ^d–m^	11.30 ^fg^	93.40 ^j–n^
Khavaran	LY/Y	1.81 ^i–o^	53.92 ^d–k^	38.40 ^j–n^	49.90 ^e–k^	13.84 ^r^	23.78 ^b–i^	4.90 ^k–q^	93.20 ^j–n^
javid	LY/Y	2.05 ^f–o^	58.62 ^c–k^	54.46 ^b–j^	54.50 ^c–k^	21.68 ^j–r^	18.47 ^d–m^	5.49 ^j–q^	165.00 ^e–j^
12	LY/P	3.08 ^c–e^	79.00 ^a–e^	68.60 ^ab^	74.80 ^a–g^	31.07 ^e–i^	27.83 ^a–d^	6.23 ^i–m^	215.20 ^b–e^
22	LY/P	3.15 ^cd^	79.70 ^a–d^	68.70 ^ab^	76.40 ^a–g^	33.90 ^d–h^	19.35 ^c–m^	8.26 ^h–j^	273.60 ^ab^
33	LY/P–Y	1.50 ^k–o^	48.61 ^g–k^	31.66 ^l–n^	45.30 ^g–k^	16.60 ^o–r^	11.72 ^l–m^	12.85 ^ef^	86.20 ^k–n^
16	W/W	3.50 ^bc^	79.80 ^a–d^	69.41 ^ab^	77.40 ^a–e^	26.13 ^g–n^	20.98 ^c–l^	3.55 ^l–q^	207.80^b–f^
35	W/W	1.42 ^m–o^	43.50 ^i–k^	28.80 ^mn^	41.00 ^h–k^	18.589 ^l–r^	14.16 ^i–m^	6.04 ^j–n^	72.40 ^mn^
39	W/W	1.38 ^no^	46.84 ^h–k^	29.65 ^mn^	39.60 ^i–k^	18.43 ^l–r^	18.83 ^d–m^	5.03 ^k–q^	80.90 ^l–n^
54	W/W	1.92 ^h–o^	54.90 ^d–k^	42.00 ^g–n^	53.20^c–k^	22.016 ^j–r^	26.87 ^a–e^	3.34 ^m–q^	96.50 ^i–n^
57	W/P	2.26 ^d–o^	66.10 ^b–j^	61.60 ^a–e^	57.70 ^c–k^	25.358 ^h–o^	29.18 ^a–c^	5.37 ^j–q^	123.20 ^g–n^
68	W/P	4.03 ^ab^	81.00 ^a–d^	71.90 ^a^	76.90 ^a–g^	30.8 ^e–i^	23.46 ^b–j^	5.50 ^j–q^	289.86 ^a^
387	W/P	2.15 ^e–o^	65.29 ^b–k^	58.00 ^a–g^	57.40 ^c–k^	25.63 ^h–o^	34.37 ^a^	8.95 ^g–i^	123.40 ^g–n^
sp 11 13	W/P–W C/P–Y C/P–Y	1.34 ^o^ 3.08 ^c–e^ 3.09 ^c–e^	38.10 ^k^ 77.72 ^a–f^ 75.10 ^a–g^	28.40 ^mn^ 66.30 ^a–c^ 65.01 ^a–d^	31.70 ^k^ 70.80 ^a–h^ 64.20 ^a–j^	15.94 ^p–r^ 34.10 ^d–g^ 26.06 ^g–n^	26.19 ^a–g^ 19.06 ^d–m^ 21.85 ^c–l^	9.84 ^gh^ 21.92 c 15.20 ^de^	64.10 ^n^ 125.60 ^g–n^ 147.01 ^e–l^
14	W/P–Y	2.76 ^c–i^	75.33 ^a–g^	65.07 ^a–d^	64.70 ^a–j^	24.41 ^i–p^	29.16 ^a–c^	5.08 ^k–q^	126.30 ^g–n^
23	W/P–Y	1.98 ^g–o^	57.00^d–k^	46.20 ^e–l^	53.30 ^c–k^	20.29 ^k–r^	20.46 ^c–l^	5.46 ^j–q^	105.70 ^h–n^
31	W/P–Y	2.16 ^d–o^	64.11 ^c–k^	59.60 ^a–f^	57.10 ^c–k^	25.54 ^h–o^	17.55 ^e–m^	3.81 ^l–q^	120.30 ^g–n^
36	W/P–Y	2.11 ^e–o^	64.23 ^c–k^	57.12 ^a–h^	56.40 ^c–k^	24.06 ^i–q^	17.57 ^e–m^	3.83 ^l–q^	213.00 ^b–e^
40	W/P–Y	2.56 ^c–j^	71.90 ^a–h^	63.30 ^a–d^	63.40 ^a–j^	36.4 ^c–f^	23.33 ^b–j^	4.86 ^k–q^	150.50 ^e–l^
Fontane	W/P–Y	2.93 ^c–g^	76.32 ^a–g^	66.00 ^a–c^	70.60 ^a–i^	35.61 ^d–f^	19.81 ^c–m^	59.73 ^a^	131.20 ^g–n^
21	W/Y	1.50 ^k–o^	51.25 ^e–k^	32.30 ^l–n^	45.50 ^g–k^	15.42 ^qr^	16.52 ^g–m^	4.42 ^l–q^	85.80 ^k–n^
27	W/Y	1.32 ^o^	41.10 ^jk^	27.90 ^n^	36.10 ^j–k^	17.71 ^n–r^	16.69 ^f–m^	10.19 ^gh^	71.90 ^mn^
28	W/Y	1.96 ^g–o^	56.60 ^d–k^	44.50 ^f–m^	53.40 ^c–k^	17.82 ^m–r^	17.53 ^e–m^	5.32 ^j–q^	108.30 ^g–n^
32	W/Y	2.63 ^c–i^	72.40 ^a–h^	63.44 ^a–d^	61.70 ^a–k^	19.76 ^k–r^	29.27 ^a–c^	6.47 ^i–l^	181.50 ^c–g^
34	W/Y	1.66 ^j–o^	50.10 ^f–k^	29.65 ^mn^	46.00 ^f–k^	14.48 ^r^	15.81 ^h–m^	8.21 ^h–j^	85.40 ^k–n^
47	W/Y	2.08 ^f–o^	61.50 ^c–k^	54.77 ^b–i^	56.30 ^c–k^	22.86 ^i–q^	32.00 ^ab^	16.88 ^d^	179.80 ^d–g^
48	W/Y	2.33 ^d–n^	66.80 ^b–j^	61.60 ^a–e^	58.90 ^a–k^	23.20 ^i–q^	18.23 ^d–m^	2.58 ^o–q^	167.33 ^e–i^
52	W/Y	1.45 ^l–o^	47.20 ^h–k^	35.90 ^k–n^	42.50 ^h–k^	16.49 ^o–r^	18.87 ^d–m^	12.95 ^ef^	77.60^l–n^
58	W/Y	4.27 ^ab^	92.64 ^ab^	70.30 ^ab^	88.60 ^ab^	28.65 ^f–j^	23.02 ^b–k^	3.36 ^l–q^	214.70 ^b–e^
67	W/Y	3.48 ^bc^	81.38 ^a–d^	69.70 ^ab^	80.90 ^a–d^	33.90 ^d–g^	21.39 ^c–l^	4.68 ^l–q^	249.70 ^a–c^

F = flesh, S = skin, W = white, LY = light yellow, Y = yellow, P = pink, TFC = Total Flavonoids Content, TPC = Total Phenolic Content, TCC = Total Carotenoid Content, TAA = Total Ascorbic Acid, TSS = Total Soluble Sugar, TSP = Total Soluble Protein. Means with different letters are significantly different at *p* ˂ 0.05 in the whole table.

**Table 2 plants-12-01707-t002:** Pearson correlation coefficient (PCC) and *p*-values link the chemical profile and antioxidant activity of the 43 breeding potato genotypes. Statistically significant pairwise comparisons are indicated. *** < 0.001.

	**TFC**	**TPC**	**DPPH**	**FRAP**	**TCC**	**TAA**	**TSS**	**TSP**
**TFC**								
**TPC**	0.945 ***							
**DPPH**	0.848 ***	0.947 ***						
**FRAP**	0.961 ***	0.972 ***	0.892 ***					
**TCC**	0.641 ***	0.712 ***	0.729 ***	0.677 ***				
**TAA**	0.083	0.099	0.150	0.093	−0.009			
**TSS**	−0.060	−0.030	−0.033	−0.018	0.101	−0.102		
**TSP**	0.837 ***	0.823 ***	0.800 ***	0.850 ***	0.546 ***	0.113	−0.165	

## Data Availability

Not applicable.

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
