# Peer review of "Antioxidant Compounds of Potato Breeding Genotypes and Commercial Cultivars with Yellow, Light Yellow, and White Flesh in Iran"

_plants, 2023, doi:10.3390/plants12081707_

Round 1
Reviewer 1 Report
Antioxidant compounds in potato breeding genotypes and commercial cultivars with yellow, light yellow, and cream flesh in Iran.
Articles of this type can be. The theme is not original. 53 potato genotypes (12 varieties and 41 hybrids) and 8 materials were studied. For me as a potato breeder, the article is relevant. It all depends on the policy of the publication.
In this article, the authors give studies of 8 compounds. In other publications, most often, only a few compounds are presented.
As regards the methods of analysing the compounds, I have no comment. In the results, 41 hybrids are analyzed, which are unstable. Specifically, I would recommend in the future: if the purpose of the study is 3 signs (yellow, light yellow and cream flesh), then for one attribute take 5 genotypes (for statistics). The results would spread to 15 varieties and analyze more clearly.
The conclusions presented are clear, although the trends are known.
The references appropriate.
The article contains 53 potato genotypes and 8 material analyzes. It would be better to give 3 genotypes to the group for statistics in the results. Now it's hard to get caught up in the abundance of results. The conclusions are not unique, because the trends are clear.
Author Response
Please check our responses in the attached file.

Reviewer 2 Report
The manuscript describes the results of a study aimed to evaluate “the concentration of phenol(s) and flavonoid(s), ascorbic acid, carotenoids, proteins, and sugars, antioxidant capaciti(es), and to analyze the relationships between the content of these chemical compounds and antioxidant activities in breeding genotypes and commercial cultivars of potatoes”. The parameters studied are quite obvious, the only positive note is the large number of genotypes.
The manuscript is very poorly written and disorganized. Many concepts are inserted into the text without a logical criterion, in a messy and repetitive way. Many sentences are incomplete, as lines 49, 388-389, 403. On the whole, little care appears to have been taken in writing the manuscript and presenting the results. The discussion of the results is confusing and not very thorough. Conclusions are inappropriate. The English is really poor and requires strong proofreading by a native speaker.
Just some concerns are added below.
Are 'cream' and 'white' the same for you?
Where did the numbers related to TPC and TFC at lines 95-96, come from? I do not understand.
In the table the genotypes should be grouped homogeneously, so as to facilitate the reading of the data. Fountains at line 310 and 'Phontaneh' in the table are the same thing?
The measure units in table must be correctly specified.
There is no correspondence between the number of clones in the table and those described in Materials and Methods (lines 315-316). This must be clarified. However, Subsection 3.1 must be summarized.
Correlation analysis is shown in subsection 2.7, together with cluster analysis and PCA. Do the correlations to lines 152-156 refer to literature data? Why is this comment here?
Who is TSF (line 263)? Where is figure S1?
References must be reported appropriately, and not with the authors' names, eg. lines 152, ,162, 354, 361-362, after the number.
Author Response

(The authors gave the same response as above.)

Reviewer 3 Report
The work is interesting, the relationships presented are interesting and coincide with the literature data. The great value of the work is that the research was carried out on a large number of genotypes and it was possible to generalize some relationships. The downside is that the research lasted only 1 year and does not take into account the influence of the environment, and this, as we know, has a large impact on the content of individual compounds. It would be good to have at least two years of results.
The conclusions are generalized, although there are too many exceptions for such generalizations to apply.
Detailed notes
line 105 - potato should be lowercase
line 110- word content is missing ( high....)
line 111 - significant effect but what, no specification
A higher content of vitamin C was found in cream potatoes with a pink skin. The earliness of the genotypes is not given, which should be taken into account, it is known that the very early and early genotypes have a higher content of vitamin C compared to the later ones.
In conclusions, the first sentence is unnecessary, it is not a conclusion.
Author Response

(The authors gave the same response as above.)

Round 2
Reviewer 2 Report
The authors made very minor changes to the manuscript which did not improve it and absolutely did not overcome all the critical issues I had highlighted for its previous version.
Author Response
Dear Reviewer,
please, see the attached file.

Round 3
Reviewer 2 Report
Some changes gave been made but these did not concern the most important critical points of the manuscript as already highlighted in the first step of the review process. This is clearly evident by the Track Changes in revised manuscript.
Author Response

(The authors gave the same response as above.)
